# The Development, Implementation, and Evaluation of a Pharmacist-Managed Therapeutic Drug Monitoring (TDM) Service for Vancomycin—A Pilot Study

**DOI:** 10.3390/pharmacy10060173

**Published:** 2022-12-12

**Authors:** Paul Firman, Ken-Soon Tan, Alexandra Clavarino, Meng-Wong Taing, Sally Dixon, Helender Singh, Karen Whitfield

**Affiliations:** 1School of Pharmacy, The University of Queensland, Woolloongabba, QLD 4102, Australia; 2School of Medicine, Griffith University, Gold Coast, QLD 4222, Australia; 3Faculty of Medicine, The University of Queensland, Herston, QLD 4006, Australia; 4School of Public Health, The University of Queensland, Saint Lucia, QLD 4072, Australia; 5Logan Hospital, Meadowbrook, QLD 4131, Australia

**Keywords:** clinical pharmacy, therapeutic drug monitoring, vancomycin, expanded scope pharmacist

## Abstract

Background: In recent years, pharmacists in Australia have been able to expand their scope to include the provision of a range of services. Although evidence has demonstrated the benefits of pharmacist-managed TDM services, recent studies have shown that these services are not prominent within Australia and that the current TDM workflow may not be optimal. Methods: An interventional pilot study was conducted of a pharmacist-managed TDM program for vancomycin at a tertiary hospital in Australia. Results: In total, 15 pharmacists participated in the program. They performed 50.5% of the medication-related pathology over the intervention period. Pharmacist involvement in the TDM process was more likely to lead to appropriate TDM sample collection (OR 87.1; 95% CI = 11.5, 661.1) and to an appropriate dose adjustment (OR 19.1; 95% CI = 1.7, 213.5). Pharmacists demonstrated increased confidence after the education and credentialling package was provided. Conclusions: This study demonstrated that a credentialling package for pharmacists can improve knowledge, skills, and confidence around the provision of pharmacist-managed TDM services for vancomycin. This may lead to the evolution of different roles and workflows enabling pharmacists to contribute more efficiently to improving medication safety and use.

## 1. Introduction

Within the healthcare system, there is increasing awareness and development of expanding the scope of pharmacists [1]. Pharmacists have been embedded within healthcare teams for many years and the research has demonstrated their impact in relation to positive patient and economic outcomes including reductions in medication errors [2,3,4], reductions in adverse effects associated with medication use, reductions in length of treatment and hospital inpatient stays, hospital avoidance including reductions in emergency department presentations and re-admissions, and more general cost savings [5,6,7,8]. Although these studies do demonstrate positive outcomes, they are generally seen in the international setting rather than locally within the Australian healthcare system.

In recent years, pharmacists in Australia have been able to expand their scope to include the provision of services such as vaccine administration (most recently the COVID-19 vaccine), partnered prescribing and point-of-care testing, for example, for HbA1c [2,3,9]. Unfortunately, current regulations in Australia prevent pharmacists from carrying out the full range of services they are clinically trained to deliver, and this limits the access patients have to these services—for example, autonomous prescribing of Schedule 4 and Schedule 8 medications and ordering and interpreting laboratory tests, especially those related to pharmacist care [10].

Therapeutic drug monitoring (TDM) is as an essential component of patient health care, and a key responsibility for pharmacists is to ensure the best possible therapeutic outcomes for patients [11]. There is minimal discussion in Australia about expanding the scope of pharmacist involvement in TDM, to include ordering medication-related pathology and optimising therapy as a result. To appropriately manage TDM, knowledge of the pharmacology, pharmacokinetics and concentration–effect relationship of the drug is required, as well as an appreciation of the laboratory assay costs [12]. Despite pharmacists possessing these skills, the extent of their participation in all aspects of the TDM pathway (from requesting the pathology to prescribing medications) or endorsed TDM Programs within hospitals and health services remains relatively minimal in Australia [10].

A recent retrospective review of TDM at one hospital in Australia (13) revealed that the current TDM workflow may not be optimal, and the role of the pharmacist in this process may not be fully utilised. The authors found that TDM samples were more likely to be taken at the appropriate time if the pharmacist had provided advice, whereas this was not the case for appropriate dose adjustment, suggesting the need for greater education and training [13].

This pilot study’s aim was to develop, implement and evaluate a pharmacist-managed TDM program, to investigate improvement in the quality of vancomycin TDM at an outer metropolitan hospital in Australia and investigate the feasibility of implementation. The program included development of a credentialling package to train pharmacists to request required pathology for vancomycin TDM and to provide full pharmacokinetic consultations and dosing recommendations to medical staff.

The objectives included:
∘An evaluation of the pharmacist-managed vancomycin TDM program by investigating improvement in the quality of vancomycin TDM;∘Development, implementation and evaluation of a credentialling package for pharmacists to enable participation in a pharmacist-managed vancomycin TDM program.

## 2. Materials and Methods

This was an interventional pilot study conducted at an outer metropolitan digital tertiary hospital in Australia. This study received full ethical approval from the institution’s ethics committee as per the National Health and Medical Research guidelines [14].

Stakeholders engaged during the study development included The Director of Pharmacy (or their delegate), Antimicrobial Stewardship Pharmacist, Medication Safety and Quality Improvement Pharmacist, a representative from the local Pathology Department, Divisional Medical Director, and nursing representatives from all the service lines (Medicine, Surgical, Women’s and Newborns), pharmacist educator, and a representative from the digital hospital program. Stakeholder engagement was used to support the program within the hospital and establish the appropriate workflow.

### 2.1. Phase 1—Development of the Vancomycin Training Package

A credentialed training package was developed specifically for vancomycin TDM. We defined credentialling as the process by which the competence of an individual could be attested (evidence-led) by a third party. The third party in this case was the researcher who is a credentialled advanced practice pharmacist. This means that the pharmacist has undergone career mapping and reflective practice portfolio creation against the Australian National Competency Standards Framework for Pharmacists [15,16,17].

The training package consisted of a 60 min face-to-face interactive lecture and evaluation consisting of both multichoice and case-based scenario style questions. The lecture included information on the pharmacokinetic, pharmacodynamic, and TDM principles of vancomycin. Information also included ways to conduct TDM in a digital hospital system, for example, ways to access and order relevant pathology and document recommendations in the patient progress notes. The training package was developed by the researcher with input and review from the stakeholders and other members of the research team. Pharmacists were required to undertake the training and then undergo a competency evaluation which consisted of 5 multiple-choice questions and 5 scenario-based cases to be deemed competent to participate in the pilot study. Pharmacists were able to access resources and local guidelines during the evaluation and were required to obtain a pass mark of 90% to be deemed competent. The evaluation was invigilated by the researcher and feedback was provided on incorrect answers.

### 2.2. Phase 2—Implementation of the Program

The 12-week program was preceded by a 4-week recruitment phase/period. The intervention group comprised of pharmacists who had successfully completed the training package and actively performed TDM whilst the control group comprised medical officers who provided usual care. Medical officers and credentialled pharmacists were able to request a vancomycin blood concentration level for therapeutic drug monitoring within the integrated electronic medical record (ieMR). Usual communication channels used within the hospital (verbal, telephone, beeper/pager system and patient progress notes) were used to communicate that a vancomycin level had been requested for a specific patient. These same communication channels were used to notify the patients’ medical team of any recommendations post-vancomycin TDM made by the credentialled pharmacist.

#### Recruitment of Pharmacists

All registered pharmacists within the pharmacy department who had at least 12 months of hospital pharmacy experience were invited to participate. Pharmacists with less than 12 months of hospital experience were excluded from participation due to lack of experience as a registered pharmacist. The research group aimed to recruit and consent a minimum of 10 pharmacists to participate in the pilot. Information about the research project was distributed to pharmacists via departmental meetings, information sharing channels (i.e., emails), and pharmacy research forums within the organisation.

### 2.3. Phase 3—Impact and Evaluation of the Program

#### 2.3.1. Vancomycin Dosing and TDM

Vancomycin dosing and TDM was evaluated using local vancomycin TDM guidelines [18]. The guidelines recommended vancomycin loading dose based on total bodyweight and maintenance dosing based on total body weight and creatine clearance (Appendix A). The guidelines recommended a trough sample to be taken approximately 48 h after initiation of therapy. A target range of 15–20 mg/L (or as guided by infectious diseases advice) was recommended. All vancomycin TDM, meeting the inclusion criteria over the study period, was collected, and included in the study using the pathology database. Data pertaining to vancomycin TDM and the outcomes of interest: appropriately timed TDM sample and appropriate dose adjustment was collected by accessing patient progress notes, medication charts and other relevant information within the ieMR. If TDM concentration measurements are to be of any significance, the timing of the blood sample collection and the interpretation of results is imperative. It is reported that most errors in interpretation of TDM results comes from an error in the time the sample was collected [5,19]. To interpret TDM properly, it must be accurately known when the TDM sample was collected with respect to when the last dose of the drug was administered. If the sample is obtained before the drug can appropriately distribute into the tissue, the concentration will be falsely elevated [5,19].

Data collected included:∘Indication for vancomycin treatment;∘Exact time of sample and relation to last dose of vancomycin (hours);∘Vancomycin dose at the time of sampling;∘Appropriateness of sampling and relevant response to results with regards to local guidelines and patient factors;∘Duration of treatment with vancomycin dose at the time of sampling;∘Dosing schedule (and any dose changes);∘Patient demographics: weight, height, renal function, length of stay and other indirect pathology measures relevant to the TDM performed;∘Pharmacist Demographics: Years registered and post-graduate qualifications;∘Data on the acceptance or rejection of the pharmacist recommendation(s) by medical officers was collected along with any variance or incorrect advice given.

#### 2.3.2. Pharmacist Satisfaction

Pharmacists were asked to complete a pre- and post-survey assessing their confidence with vancomycin TDM. This survey can be found in Appendix B and consisted of 11 questions using a 5-point Likert scale (1 = strongly disagree, 2 = disagree, 3 = neutral, 4 = agree and 5 = strongly agree). The survey was developed from a locally used and validated confidence tool used for performance development. The survey was piloted using five pharmacists not involved in the pilot; based on their feedback, minor changes were incorporated in the final survey.

#### 2.3.3. Data Analysis

All analyses were performed using the Stata statistical software package (Version 15) [20].

i.Outcomes of Interest: appropriate sample collection and appropriate dose adjustment

Mixed effects logistic regression modelling was used to assess the effects of pharmacist involvement, patient age and patient sex on appropriate sample collection, incorporating a patient-level random effect. Mixed effects logistic regression modelling was also used to assess the effects of pharmacist involvement, dose adjustment, age, and sex on appropriate dose adjustment, incorporating a patient-level random effect.

ii.Pre- and Post-Credentialling Confidence

The median, interquartile range (IQR) and *p*-value were used to compare pre- and post-credentialling confidence ratings measured using a 5-point ordinal Likert scale (1 = strongly disagree, 2 = disagree, 3 = neutral, 4 = agree and 5 = strongly agree). The significance level was set at *p* < 0.053.

## 3. Results

A total of 103 vancomycin TDM observations from 40 patients (Figure 1) were made during the study period. Of these, 52 (50.5%) were in the intervention arm (pharmacist requested pathology). The patients had a mean age of 48.7 years and were predominately male (63%) (Table 1). Of the 52 pharmacist observations, medical officers agreed with the pharmacist dosage recommendations 100% of the time and made the subsequent dosage change as recommended.

A total of 15 pharmacists were recruited to the study. From Table 2, 80% of the participants were female with the average years registered being 9.1 years and 53.3% had a postgraduate qualification (a postgraduate qualification is obtained by a student who has successfully completed an undergraduate degree level course at a college or university and has then undertaken further study at a more advanced level). All 15 pharmacists successfully completed the requirement to be credentialled (90% or greater in the competency evaluation) on the first attempt.

Pharmacist involvement in the TDM process saw 98% of the TDM samples collected at the appropriate time as per local guidelines compared to the medical officer control group (98.0% vs. 34.6%). Overall, the pharmacist involvement was more likely to lead to appropriately timed TDM sampling (OR 87.1; 95% CI = 11.5, 661.1); see Table 3.

After the pharmacist recommendations, therapeutic targets (15–20 mg/L) were achieved 49.1% of the times compared to medical officers where therapeutic targets (15–20 mg/L) where achieved (38.5%).

Medical officers calculated and prescribed all loading doses and initial maintenance doses in the study. Medical officers calculated 81.8% of loading doses and 84% of maintenance doses in line with the local guidelines.

Pharmacist involvement in the TDM process led to more appropriate maintenance dosing and adjustments being recommended compared to the control group (96.1% vs. 75.8%) (Table 4).

The involvement of the pharmacist saw the documented communication of vancomycin therapy within the patient medical record increase with pharmacist involvement (100.0% vs. 51.9%) (Table 5).

### Pharmacist Confidence Questionnaire

Questionnaires were completed before and after completing the credentialling program. All questions showed positive improvements on the items presented (Table 6), with participants feeling more confident on all aspects of vancomycin therapeutic drug monitoring. Data for the variables pre-credentialling and post-credentialling were not normally distributed, therefore Wilcoxon signed-rank tests were performed for each questionnaire item to test whether post-credentialling ratings were greater than pre-credentialling ratings. *p*-values < 0.05 were considered statistically significant. Results for all questions except questions three and five were statistically significant, demonstrating that post-credentialling confidence ratings were greater than pre-credentialling ratings for all questionnaire items except items three and five.

## 4. Discussion

This study developed, implemented, and evaluated the feasibility of a pilot pharmacist-managed vancomycin TDM program at an outer metropolitan digital hospital. The results of this pilot program suggested that expanding the involvement of pharmacists within the TDM process that included ordering the required pathology significantly improved the likelihood of appropriate sample collection and dose adjustment for vancomycin TDM. The study also indicated that pharmacists felt more confident after completion of a vancomycin TDM credentialling program and participation in a pharmacist-managed TDM program.

Clinical pharmacokinetic monitoring is an integral part of pharmaceutical care for particular patients based on their specific pharmacotherapy, disease states and related factors and treatment goals [11,21]. Pharmacists have a key role to play in this area, particularly with their knowledge of the pharmacokinetic and pharmacodynamic properties of medications, but this has not been fully realised within Australia as is the case in other countries such as the USA, UK and Japan [8,10,21,22]. Studies have also shown benefit from pharmacokinetic dosing services with aminoglycosides and vancomycin [23,24,25,26]. This current study has demonstrated the benefits of pharmacists undertaking this increased role at one site within the Australian healthcare system. Within this study, pharmacists were able to order pathology and make recommendations for vancomycin TDM after completion of a credentialling package. Over the study period, pharmacists requested 50.5% of vancomycin TDM with medical officers agreeing with their recommendations around vancomycin dosing 100% of the time.

Results from the study indicated that having a pharmacist order the required pathology for vancomycin TDM increased the odds of an appropriate sample (OR = 87.1). These findings build on our previous findings which showed that samples for TDM were more likely to be appropriate if the pharmacist had provided relevant written advice (i.e., documenting when a pathology sample was required for the medical officer) (OR = 2.0; 95% CI = 1.4–2.9) [13]. Here, we have demonstrated further improvement with greater pharmacist involvement. This impact on appropriate sampling may allow better review of the pathology result and medication leading to more appropriate and safer use of this medication.

When interpreting TDM results, factors such as the time the TDM sample were considered, as well as the time when the last dose was administered and the patient’s response to the therapy. This information can be used to identify the most appropriate dosage regimen to achieve the optimal response with minimal toxicity. All of these factors form a part of clinical pharmacist activities [27]. Pharmacist involvement in the TDM process led to more appropriate maintenance dosing and adjustments being recommended compared to the control group (96.1% vs. 75.8%). This study demonstrated that having the credentialled pharmacists in TDM take a proactive role in providing a pharmacokinetic consultations on the results of vancomycin TDM improves the likelihood of having an appropriate dose adjustment (OR = 19.1). A similar study evaluating the effect of a pharmacist-directed vancomycin dosing and monitoring pilot program in the US by Marquis et al. showed similar positive results with the percentage of patients who received optimal vancomycin dosing being significantly higher in the postimplementation group (96.8% vs. 40.4%, *p* < 0.001) [24].

The pharmacy profession has undergone remarkable change in recent years. Pharmacist roles have expanded traditional boundaries to encompass more patient-centered services [28]. Despite this, pharmacists anecdotally reported little confidence in their clinical decision-making skills [29,30]. The recent literature has discussed that barriers may include hierarchy of the medical system, role definitions, evolution of responsibility, ownership of decisions for confidence building, quality and consequences of mentorship and personality traits upon admission to the pharmacy profession [29,31]. This pharmacist-managed vancomycin TDM pilot program required a pharmacist to undergo a credentialling package to participate within the program and equip them with the skills and confidence required to participate in the pilot in what could be described as a non-traditional role for the pharmacist. Following participation in the training program, the pharmacists reported increased confidence in performing the required duties. The moving of pharmacists into this non-traditional role was supported by the medical officers within the facility with 100% of recommendations accepted by the treating team. This rate is similar to that reported in a study in Japan (2016), which utilised pharmacist feedback to medical officers on vancomycin TDM. This study showed that acceptance rates for pharmacist recommendations to increase the dose of vancomycin (70%), decrease the vancomycin dose (94%) and to change to an alternative agent were 100% [32].

This model of vancomycin TDM credentialling represents a program that is suitable for application to other hospitals within Australia. The study showed the impact pharmacists can have on patient care and the safe use of high-risk medication. The ability to have a structured credentialling program provides credibility to the service and the important role TDM is in medication management. It can also have a positive impact on patient outcomes with multifaceted interventions including educational material on vancomycin dosing, monitoring and nephrotoxicity and guideline implementation. Moving into the future, standardised policy of the role at a local, state, or national level with standardised training could be a solution for this role of the pharmacists being more prominent within the Australian Hospital and Health System. This study and the implementation of a pharmacist-managed vancomycin TDM program also must consider research and consensus guideline updates that have occurred since the pilot was trialled. There is currently minimal to no data on the safety and efficacy of targeted trough concentrations that are currently used for vancomycin TDM. A major concern with vancomycin is the occurrence of acute kidney injury (AKI) and, therefore, based on current evidence and given the narrow vancomycin area under the curve (AUC) range for therapeutic effect and minimal AKI risk, the most accurate and optimal way to manage vancomycin dosing should be through AUC-guided dosing and monitoring using Bayesian software programs. Pharmacists are best placed for this type of TDM, but it is realistic to assume that not all hospital and health services are able to set up this type of service especially utilising Bayesian-derived AUC monitoring, and consideration must be made for this [33].

### Limitations

Although this pilot study demonstrates promising results and positive outcomes, several limitations are outlined. This is a pilot study, conducted at one site with one medication and the generalisability of the results and program make it difficult to draw conclusions for implementation at other institutions. Further large-scale studies and feedback from key stakeholders is required to investigate feasibility and impact in the future. In addition, as mentioned, current recommendations for vancomycin TDM are provided through AUC-guided dosing utilising Bayesian dosing software which the pilot institution was not equipped for. Local guidelines recommending trough levels were utilised instead.

## 5. Conclusions

The role of pharmacists in measuring, monitoring, and managing medication use in hospitals and health systems continues to be significant, important, and growing. Although this study demonstrated that a credentialling package for pharmacists can improve knowledge, skills, and confidence around the provision of pharmacist-managed TDM services for vancomycin and can improve the quality of vancomycin TDM within the hospital, further larger scale studies and feedback from key stakeholders (medical officers and nursing staff) are required to fully understand the feasibility and impact. This may lead to the evolution of different roles and workflows enabling pharmacists to contribute more efficiently to improving medication safety and use.

## Figures and Tables

**Figure 1 pharmacy-10-00173-f001:**
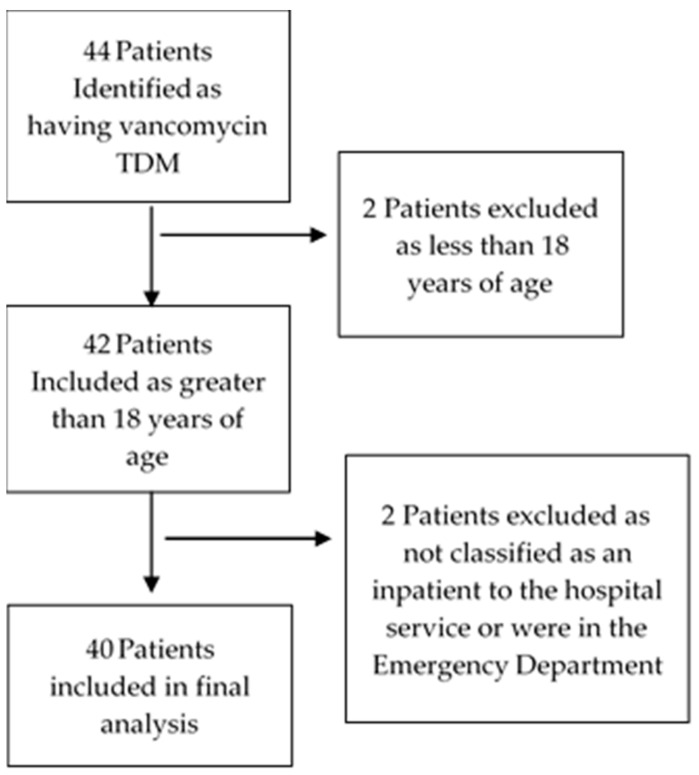
Patient Selection.

**Table 1 pharmacy-10-00173-t001:** Patient Demographics.

Variable	N = 40	
	Median	SD
Age (years)	44.0 (19–90)	17.83
Weight (kilograms)	82.0 (52.0–167.2)	26.9
Height (centimetres)	175 (167–180)	5.2
Creatinine (µmole/L) (Reference Range 45–90 µmol/L)	79 (41–657)	124.4
*Sex*		
Male	25 (63%)	
Female	15 (37%)	

**Table 2 pharmacy-10-00173-t002:** Pharmacist Demographics.

Variable	N = 15
*Sex*	
Female	12 (80%)
Male	3 (20%)
Years Registered (mean)	9.1 years (1–20)
*Postgraduate Qualification*	
No qualification	7 (46.7%)
Graduate Certificate	2 (13.3%)
Graduate Diploma	4 (26.7%)
Master’s degree	2 (13.3%)

**Table 3 pharmacy-10-00173-t003:** TDM sample timing (Medical officers and Pharmacist) (*n* = 40 patients, 103 observations).

Outcome: Appropriately Timed TDM Sample		
**Variable**	**Observations (N)**	**Percentage (%)**
Medical Officer (Control Group)	18/52	34.6%
Pharmacist	50/51	98.0%
**Variable**	**Adjusted Odds Ratio ^#^**	**95% CI**
Pharmacist Involvement ^#^		
Yes	87.1 *	(11.5, 661.1)
No	Ref	Ref

* *p* < 0.001; ^#^ Calculated by mixed effects logistic regression analysis.

**Table 4 pharmacy-10-00173-t004:** Vancomycin dose adjustments (Medical officers vs Pharmacist (*n* = 40 patients, 103 observations).

Outcome: Appropriate Maintenance Dose Calculation and Adjustment (Maintenance Dose)		
**Variable**	**Observations (N)**	**Percentage (%)**
Medical Officer (Control Group)	22/29	75.8%
Pharmacist	49/51	96.1%
**Variable**	**Adjusted Odds Ratio ^#^**	**95% CI**
Pharmacist Involvement ^#^		
Yes	19.1 *	(1.7, 213.5)
No	Ref	Ref

* *p* < 0.001; ^#^ Calculated by mixed effects logistic regression analysis.

**Table 5 pharmacy-10-00173-t005:** Results of appropriate documentation of TDM information in patient medical records.

Outcome: Appropriate Documentation for TDM		
**Variable**	**Observations (N)**	**Percentage (%)**
Medical Officer (Control Group)	27/52	51.9%
Pharmacist	51/51	100.0%

**Table 6 pharmacy-10-00173-t006:** Pre- and post-credentialling package confidence questionnaire.

	Post-Credentialling Confidence Rating Greater than Pre-Credentialling Rating
	*p*-Value ^a^
1. I feel confident providing advice on vancomycin therapeutic drug monitoring	0.01
2. I feel confident about when to recommend an initial drug measurement level for vancomycin therapeutic drug	<0.001
3. I do not feel confident about when to recommend subsequent drug measurement levels for vancomycin	0.10
4. I am unsure which infections vancomycin is used to treat	<0.001
5. I feel unsure about providing advice on vancomycin dosing and monitoring in obese patients	0.10
6. I feel confident in providing advice on vancomycin dosing and monitoring in renal patients	0.01
7. I feel confident in recommending vancomycin loading doses	0.01
8. I feel confident in recommending vancomycin maintenance doses	0.01
9. I am familiar with the guidelines and policies around vancomycin and vancomycin	0.01
10. I actively provide advice to the multidisciplinary team about Vancomycin Therapeutic Drug Monitoring	0.02
11. I feel the multidisciplinary team are receptive of my advice provided for vancomycin therapeutic drug monitoring	0.04

Scored using a 5-point ordinal Likert scale (1 = strongly disagree, 2 = disagree, 3 = neutral, 4 = agree and 5 = strongly agree)**;**
^a^ Wilcoxon signed-ranks test.

## Data Availability

The data presented in this study are available on request from the corresponding author. The data are not publicly available due to privacy.

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
