# Peer review of "The Development, Implementation, and Evaluation of a Pharmacist-Managed Therapeutic Drug Monitoring (TDM) Service for Vancomycin—A Pilot Study"

_pharmacy, 2022, doi:10.3390/pharmacy10060173_

Round 1

Reviewer 1 Report

The authors address a very important issue. We definitely agree that pharmacists should expand their scope including TDM in Australia as well as in other countries. Pharmacists can make an essential contribution in TDM.

However, I would like to question whether the focus of the study should be on pharmacist satisfaction rather than on the quality improvement as it is implied by the abstract. There is many information about participating pharmacists, but collected data for TDM analysis e.g. dose, time of sampling, patient demographics, are missing. The approach of dose adaption (AUC?) as well as target are missing in the method section. Furthermore, sampling time and documentation of TDM are important surrogate parameters after the implementation of a pharmacist managed Therapeutic Drug Monitoring, but also target attainment would be interesting. These parameters should be described more in detail. Overall, patient number in relation to the number of pharmacists included seems very low. Due to the low information content, I do not consider the manuscript  in its current form suitable for publication.

Author Response

Good Afternoon, 

We thank the reviewers for the comments and feedback and have addressed them as per the attached document. 

Many Thanks

Paul Firman

Reviewer 2 Report

Pharmacist Confidence Questionnaire

I don't understand the purpose of this survey; shouldn't an objective evaluation of the appropriateness of TDM be sufficient? I also wonder if this is just an evaluation of the training that was done in the pilot study.

The "control group consisted of healthcare professionals who provide usual medical care," but I don't think there is any mention of this group.

Also, as an observational subject, I think it would be desirable to evaluate the appropriateness of the training, preferably by evaluating the TDM consultations of "trained pharmacists" and "non-trained pharmacists".

If this paper is an examination of the development and superiority of pharmacists' TDM intervention systems in healthcare settings, I believe that the final evaluation should be made by physicians, nurses, and other healthcare professionals to assess the superiority of pharmacist intervention

Author Response

(The authors gave the same response as above.)

Round 2

Reviewer 1 Report

I think the focus of the manuscript still needs improvement. The aim of the study should  be first and foremost to improve the quality of vancomycin TDM as implied in title and abstract, that is, is the pharmacist improving vancomycin TDM and thus achieving requested target levels. Satisfaction might be a second aspect.

The authors present the results whether the correct dosage, time, documentation as comparison between groups, but not the raw data. We do not recommend to include this tables with logistic regression analysis. We still miss actual dose used in this patients, time of sampling, vancomycin concentration. In demographic data essential parameters to estimate the pharmacokinetics in this group e.g. kidney function, height, weight is still missing.

I think vancomycin concentration has to be already part of this project. How can you decide, if dose adjustment was appropriate, if TDM is improved by pharmacists, if you don’t evaluate the vancomycin concentration. This data should already be available and therefore be described in the paper as evidence that pharmacists improve TDM.

Author Response

Thank you for the reviewers comments. The authors have addressed the comments in the attached document.

Many Thanks

Paul Firman

Reviewer 2 Report

No additional comments.

Author Response

Paul Firman
The University of Queensland – School of Pharmacy
20 Cornwall Street, Woollongabba

Queensland. 4102

Australia

 30th November 2022

Dear Ms Mia Yu

I wish to submit feedback and updated manuscript following the reviewer’s comments, for an original research article titled “The development, implementation, and evaluation of a pharmacist managed Therapeutic Drug Monitoring (TDM) service for vancomycin – A pilot study” for consideration by Pharmacy within the special issue Pharmacy: State-of-the-Art and perspectives in Australia

We thank the reviewers for the comments and feedback and have addressed them as follows

Reviewer Two

No additional comments

We thank you for your feedback on the manuscript

No update to manuscript

We have no conflicts of interest to disclose.

Please address all correspondence concerning this manuscript to me at paul.firman@health.qld.gov.au

Thank you for your consideration of this manuscript.

Sincerely,

Paul Firman
